# Polarization is the psychological foundation of collective engagement
Laura G. E. Smith [1,5] ✉, Emma F. Thomas [2,5], Ana-Maria Bliuc [3,5] & Craig McGarty[4]

The term polarization is used to describe both the division of a society into opposing groups (political polarization), and a social psychological phenomenon (group polarization) whereby people adopt more extreme positions after discussion. We explain how group polarization underpins the political polarization phenomenon: Social interaction, for example through social media, enables groups to form in such a way that their beliefs about what should be done to change the world—and how this differs from the stance of other groups—become integrated as aspects of a new, shared social identity. This provides a basis for mobilization to collective action.

In political and social science, the term polarization is often used to describe the division of society into opposing groups that have become distant in their positions. Political polarization occurs when this process is applied to opposing groups based on identification with a specific party[1] (e.g., Democrat versus Republican), position[2] (e.g., climate believer or sceptic) or policy[3]. Political polarization attracts much attention because it has been linked to the rise of hostility and conflict in politics. Thus, violent clashes between supporters and opponents of immigrants at the port of Calais[4]; intransigence stemming from staunch divides between those who support versus oppose action on climate change[5], or the British Exit from the European Union (Brexit)[6]; and the political violence that ensued from the January 2021 US Capitol Insurrection[7] are seen as outcomes of political polarization. These examples suggest that polarization can engender extreme hostility, conflict and (occasionally) intergroup violence. Some commentators suggest that such processes diminish our humanity, and therefore, "the more polarized we become; the weaker humanity becomes"[8]; others are concerned that these pose existential threats to democracy and democratic institutions.

We propose an alternative, perhaps more optimistic, view of polarization. While we accept that intergroup conflict and hostility are possible outcomes of polarization, these are not the only outcomes. There is a risk that, just as Le Bon[9] put forward an influential yet flawed analysis of crowd behaviour in the late 19th Century[10,11], here too there has been inadequate recognition of the broader impacts of the social and psychological processes that allow polarization to emerge. Alongside an age of polarization, we are also living in an age of protest[12]. As we argue, social psychological processes of polarization that have driven people to extreme forms of hostility and conflict have also been influential in producing mass movements to challenge the inequalities experienced by women, or people of colour, and overturning dictatorships. Far from diminishing humanity, these movements have elevated human rights and access. If one only studies the

outcomes of polarization—such as revolution and social change on one hand, or intergroup hostility and conflict on the other, one only sees one part of the story, and not the psychological processes that are common to these disparate outcomes. Thus, one consequence of a broader contextualization of the origins and effects of polarization is a failure to recognize the important role that polarization plays in fostering political engagement, which is itself critical for pluralistic societies in general, and democracies in particular.

In this Perspective article, we bring together the above observations to build a more balanced view of the origins, nature, and effects of polarization. Our starting point is the observation that, in psychological science the term polarization has a long-standing meaning: the tendency for moderate attitudes to become more extreme after discussion. This effect was isolated in the laboratory by Moscovici and Zavalloni, who labelled it group polarization[13]. It is illustrated in Fig. 1a, b. Group polarization is easy to reproduce, both in the laboratory[14], and in everyday life[13,15]. Moscovici and Zavalloni's experiments showed that Stoner's idea of risky shift[16] was a more specific case of the general tendency for groups to shift to a more extreme version of the initial tendency within the group. It's important to note at the outset that attitudinal extremity (attitudinal extremism) is not the same as support for violent extremism, although the two are often conflated[17], a point we return to below.

## The need to reconceptualize polarization

The scientific process of isolating the group polarization phenomenon dates back to Lewin, who demonstrated that people were more likely to stick with crucial decisions to make sacrifices support the Second World War effort when those decisions were made in interactive groups than when those decisions were made by isolated individuals[18,19]. Group polarization and the earlier findings by Lewin provided psychology with ways to understand the role of communication within groups in producing social change,

¹University of Bath, Bath, UK. ²Flinders University, Adelaide, Australia. ³University of Dundee, Dundee, UK. ⁴Western Sydney University, Sydney, Australia. ⁵These authors contributed equally: Laura G. E. Smith, Emma F. Thomas, Ana-Maria Bliuc. ✉e-mail: l.g.e.smith@bath.ac.uk

**Fig. 1 | The classic group polarization effect, manifesting through active differentiation between groups as political polarization. a** Before discussion, a group made of up four people (as represented by the red letters: A, B, C, and D) who all have moderate anti positions on an issue. **b** After discussing the issue, the group members shift to a more extreme anti position. **c** Before discussion, a group made up of 8 people, four of whom start with a moderate anti position (the red letters representing four people: A, B, C, D), and four of whom start with a moderate pro position (the blue letters representing four people: E, F, G, H). **d** After discussion, the 8 people split (polarize) into two groups, with separate groups indicated by the red and blue letters, respectively.

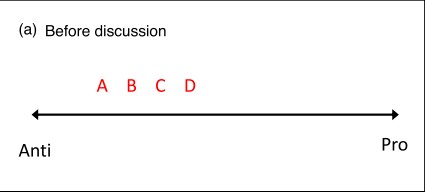

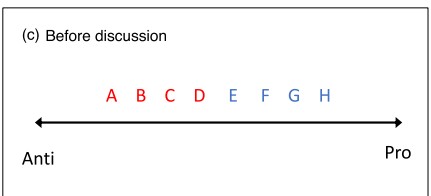

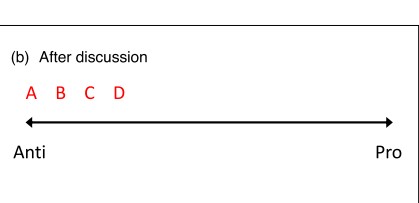

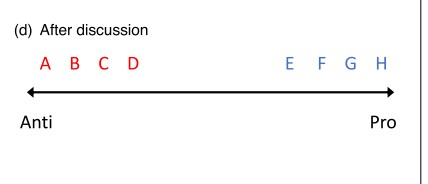

including in ways that are problematic. For example, when racially prejudiced school children discussed their views in groups, they became more prejudiced[15].

Three-quarters of a century after the research on group polarization by Moscovici and Zavalloni, a fresh look at group polarization is needed for three reasons. The first and most obvious reason is that the concept of group polarization is relevant to explaining political polarization, as Fig. 1 suggests. Traditional group polarization involves one group becoming more extreme in one direction (Fig. 1a, b), but if we imagine two groups polarizing in opposite directions, then we have the psychological foundations of political polarization (Fig. 1c, d). Indeed, Doise showed that group polarization increased in the presence of a rival group with opposing views[20] (see Fig. 1c, d for polarization involving two small groups). Thus, research on the psychological foundations of political polarization can be traced back to the earliest work on group polarization but the connections between the two have not yet been fully articulated.

The second reason to look again at group polarization is because it requires communication, and online technologies have created new means for people to communicate, in ways that are increasingly linked to social change[21–23]. Indeed, debates about political polarization have emerged in the context of rapid, dramatic technological change. Social networking services (SNSs) such as those operated by Meta and X, but also Telegram, Reddit, Gab, and many others, enable people to associate with likeminded others from anywhere in the world (even if some of those others are not real people but automated bots), engage in discussions that help them validate their views and form consensus; confront people with opposing views, and express views (often under the cover of anonymity) that bypass political, cultural, and legal restrictions on communication. It is this basic process of interaction, discussion and debate that, we contend, underpins polarization and protest, engagement and political violence. For many years, the group polarization phenomenon attracted little scholarly attention[24], however, it has recently been experiencing a renaissance, catalysed by the affordances of Web 2.0 and now Web3 for social interaction and connectivity[25]. Social media, large language models, generative AI, GPT-3 and GPT-4[26], and applications such as ChatGPT[27], have provided the capacity to generate and spread communications on a massive scale. While some of these technologies can lower the barriers to participation[28,29] and thus equip people to catalyse movements to achieve greater rights and access for disadvantaged people, it has also been noted by the United Nations (among others) that they provide the potential for widespread social and political harm[30].

The third reason for a new look at polarization relates to the group part of group polarization. The group polarization phenomenon was isolated just before a massive expansion in the study of group processes derived from the social identity perspective[31,32]. More than fifty years after the paper by Moscovici and Zavalloni[13], we now know much more about such identities, and how those group identities form and impact on attitudes and behaviours. While important steps have been taken to link that knowledge with

the concept of group polarization in the work of Postmes, Haslam and Swaab[33] and others, the connection between the identity processes involved in group polarization[34,35] and political polarization is, as yet, weakly articulated. As we will argue below, processes of group polarization are fundamental to group formation, group life, mobilization, and social change, and part of social change involves forming groups based on opinions[36] about changing the world[37].

Before we can elaborate and extend our arguments, we first need to be clear about definitions. In popular discourse and in social science, the term polarization has multiple and divergent meanings[25,38] (see Box 1, for some non-exhaustive examples). This has partly arisen to accommodate how communication technologies have afforded new ways for people to communicate (and polarize), and partly because of the related opportunities for new forms of data and new methods to be used in polarization research. Some of these definitions combine polarized outcomes or constructs (such as attitudes, opinions, or emotions), with descriptions of temporal processes of polarization (e.g., group discussion). Because polarization research now includes this diverse set of concepts, there is a lack of empirical clarity in polarization research[25,38–40]. Furthermore, this diversity of definitions and operationalizations of (broadly termed) polarization phenomena risks foreclosing theoretical integration and advancement. In turn, there is limited understanding in the field of the common psychological mechanisms underlying, and downstream consequences of, those phenomena—and how they connect.

Moreover, as Jost and colleagues have explained, different types of polarization can affect each other[40]. For instance, issue-based polarization is often a driver of affective polarization[41], as in cases where dissent over a newly emerged issue in society (e.g., the COVID-19 pandemic) polarizes the public into opposing groups, which then develop hostile attitudes towards each other (affective polarization). In this sense, political polarization is best conceptualized as a process incorporating both issue and affective polarization, with issue-driven dissent creating conditions for issue-driven polarization, while affective polarization being its manifestation (or the attitude - behavioural component within this process[42]). Similarly, the interconnections between different forms of polarization are clear in cases where collection action occurs in response to opposing groups becoming more extreme through exposure to each other's ideology and actions (co-radicalization; Box 1). Clearer theorizing about these phenomena that disentangles processes from outcomes could help to explain why there has been such diversity in findings, definitions, and operationalizations of polarization.

## Political polarization as active differentiation between opinion-based groups

We argue that group polarization—referring to the processes of social interaction that lead to more extreme opinions within a group—can cause political polarization: the divergence of two groups on opinion-based lines

in relation to each other. This extremization that occurs through group polarization indicates a shift towards a more extreme end of a continuum of beliefs, attitudes[43], or opinions, as demonstrated via the group polarization studies[13,15]. Groups with opposing polarized opinions differentiate themselves from each other, causing political polarization (Fig. 1d). In this way, group polarization has the potential to bring people together to engage in collective action. Any group can undergo this process of extremization via polarization, but this does not mean that they are extremists[17], or that they support the use of violence. The nature of collective action that follows these processes depends upon the social context and the norms that form within those groups[21], and can vary from benevolent support, conventional activism, to political violence. Thus, such collective action can have a range of impacts, from progress to conflict and harm.

We summarize our arguments in the six propositions in Box 2. The connection between the propositions is visualized in Fig. 2. One caveat is that we do not suggest that the propositions are linear in the sense of uniform or sequential stages or steps. Rather, our goal is to clarify the psychological processes that need to occur for the processes of group polarization to lead to political polarization and mobilization to collective action. Together, the propositions provide a framework for understanding how group polarization and political polarization connect to each other, with (positive and negative) consequences for society at large. First, we explain the background to these ideas, and then we illustrate the processes referred to in those propositions with reference to three examples that have global significance.

## Social psychological phenomena emerge from social interaction

When two or more people interact, they mutually influence each other. Such social influence should be central to the study of many of the social sciences[44]. However, there had been a sharp decline in the study of live or real-time interaction in the period from the 1960s[24]. Part of the problem was that the study of groups of interacting people is both more expensive and far more complex in statistical terms than the analysis of individuals. Statistical tests such as linear regressions and analyses of variance assume that participants' observations are independent of each other, that is, that individuals do not change or affect each other's data (the assumption of independence of observations), and therefore cannot be used if participants interact in groups

---

## Box 1 | Common definitions and conceptualizations of polarization phenomena

| Term | Definition |
|---|---|
| Individual polarization | The tendency for individuals to become more extreme in their own attitudes, opinions, thinking[149], communications[38], or in the strength of their identification with a group[150]. |
| Group polarization[13,15] | After group discussion, a group's position or behaviour becomes more extreme but in the same direction as the average of the group members' initial opinions on the issue. |
| Intergroup polarization; political polarization | Two or more groups (or political parties) diverge or become more partisan in relation to each other[1,151] (linked to affective and issue-based polarization). |
| Affective polarization[4,6,85] | Members of different social groups (or political parties) have increasingly negative feelings or attitudes towards outgroup/s[5,151]. |
| Issue-based polarization | Divergence in positions based on support for contrasting policies or issues (e.g., issue or ideological distancing[41]). It can occur between groups or within the same group (e.g., factions within a political party supporting different policies). |
| Co-radicalization[132] | A group becomes more extreme in reaction to the rhetoric and actions of another group, and that other group uses the rhetoric and actions of the first group to increase support for their agenda and justify more extreme intergroup actions and beliefs. |

---

## Box 2 | Propositions for the connection between group polarization, political polarization, and collective action

| Propositions |
|---|
| 1. Communication makes it possible for people to connect with like-minded others and to polarize, consensualize and collectivize over social issues[23,37,77,78,81,86,89]. Lack of communication therefore forecloses the opportunity for new groups to form through group polarization. |
| 2. People form opinions about how the world should be, and unite in groups with people who share those opinions, in order to achieve desired changes[2,36,37,147]. |
| 3. The new groups (described in #2) form when people align and integrate emotion, beliefs, and ideas for actions through communication and internalize them as aspects of a shared social identity[37,81]. These new social identities (opinion-based groups) are therefore the result of processes of *group polarization*. |
| 4. The creation of new opinion-based groups via *group polarization* (as per #2)—who articulate their own position in terms of how it differs to that of an opposing group—leads to *political polarization*. |
| 5. Powerful, opposing groups will try to subvert, marginalize, or prevent the communication referred to in #1 and 2[60,148]. |
| 6. Collective action can be an outcome of the processes articulated in #1, 2, and 4. Social change will tend to rest on grassroots solutions[147], that involve people agreeing how the world can be other than it is[37], and seeing themselves to be part of the movement to change it[147]. |

during a study[24,45]. Yet, in considering groups of people who are interlinked, the interdependence—or, mutual influence—that results from those interactions is precisely what we are interested in finding out about. Kashy and Kenny argued that, "Social psychologists should treat interdependence not as a statistical nuisance, but rather as an important social psychological phenomenon that should be studied," (p. 452)[46].

Since the 1960s, there have been cogent efforts in psychology to meet the challenge of studying social interaction while addressing the factors that might give rise to political polarization. One very well-known approach is social impact theory, which argues that the social influence of people on others is driven by the combination of three factors: the strength of the sources of influence, the immediacy of those sources, and the number of sources[47]. According to the theory, you are more likely to be influenced when many people, who are linked to you, argue strongly for their position. This idea was generalized and extended in dynamic social impact theory[48], and in the bubble theory of rapid social change[49]. Imagine an array of people connected to each other across distances in an online network and that those people hold either a majority or minority position and who engage in repeated interactions over time. Agent-based modelling simulations show that it results in a consistent pattern of organization and clustering into divergent groups[50,51]. In dynamic social impact theory, the majority and minority positions are seen as equally attractive (thus explaining why minority positions can persist in society), but bias introduced, for example by the socio-political climate and specific trends, can help the minority position to become dominant[49].

The question emerges, however, as to whether the dynamics of social interaction underpinning support for divergent positions are enough for full-scale political polarization (as would be implied by dynamic social impact theory[48])? If they are, then political polarization would be an inevitable outcome of free social interaction. Put another way, if we merely allow people to talk to each other then, given enough time, we will end up with divided nations and communities[41]. It is certainly true that SNSs allow, on a massive scale (in the form illustrated in Fig. 3), the formation of so-called echo chambers or filter bubbles where people encounter few contrary views[51,52]. Some SNSs use algorithms that ensure that users tend to encounter similar views to their own[53] (or, of course, just views that have been promoted by advertising). Such echo chambers can be seen in political arguments and in debates about science and pseudo-science[52].

However, algorithmic feeds and echo chambers of likeminded individuals do not necessarily lead to political polarization[54,55]. Indeed, work by Barbera and colleagues encourages a more nuanced view. Using a novel method for estimating ideological positions on Twitter, these authors showed that people holding the liberal and conservative ideological positions did seem to form coherent echo chambers (so that liberals tweeted to liberals and conservatives to conservatives) on some issues and at some times, but not in other circumstances[56]. Discussion of the US presidential election showed the formation of echo chambers, but discussion of the Superbowl did not[56]. Critically for our purposes, this suggests that echo chambers were linked to the formation, and transformation, of groups that expressed ideological positions (with some important subtleties). In other words, Barbera et al.'s results suggest that echo chambers are not just emergent outcomes of discussions in the way that we would expect from dynamic social impact theory. As their research shows, on Twitter, people tend to polarize and communicate predominantly with ingroup members when the topic of conversation is aligned to their salient political identities, and when this allows them to express their ideological views about the ideal arrangement of society[57]. These observations are consistent with the social identity approach—that the groups that we stand with (versus those whom we are against) are those with whom we feel we belong, but also those groups that help us to best make sense of the social context that confronts us[32,58].

## Communication makes it possible for people to connect with like-minded others

Group polarization occurs through communication (Proposition 1)—whether online or offline. Therefore, while SNSs provide opportunities for polarization, we note that SNSs are not necessary for polarization to occur. It is communication, rather than the affordances of SNSs, that is necessary for people to collectivize over social issues. This is because communication is the

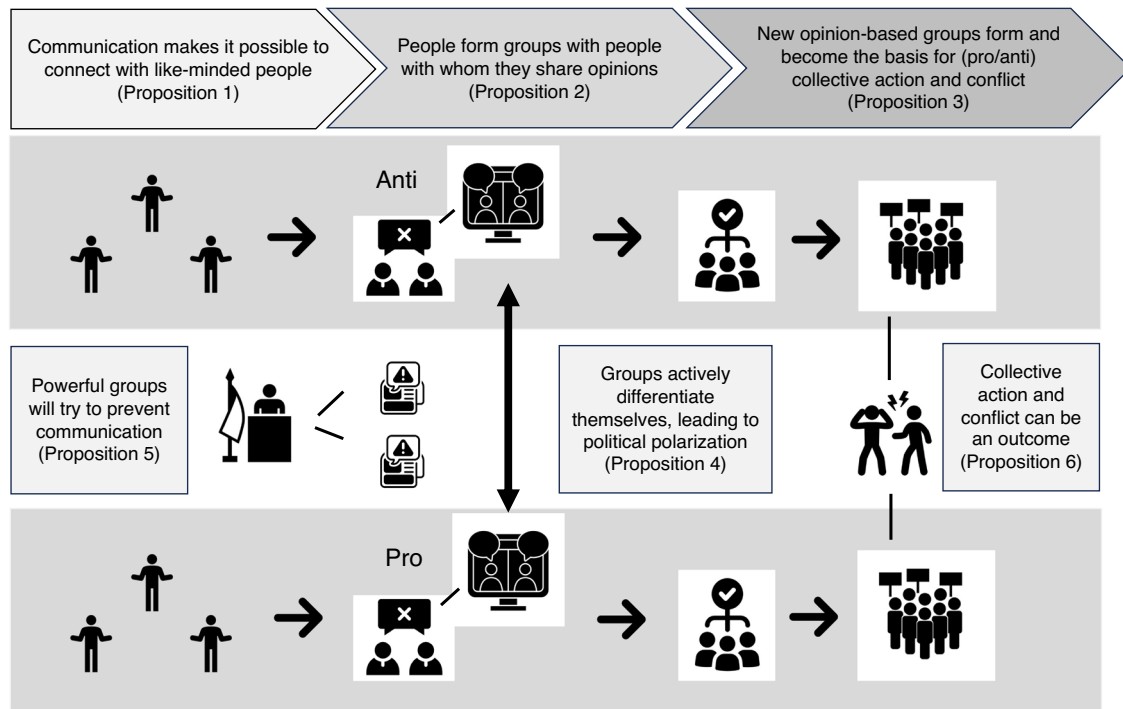

**Fig. 2 | How group polarization leads to collective action and conflict via political polarization.** This figure illustrates connections between the propositions in Box 2. When people communicate with likeminded others, they can form groups based upon opinions. Groups based on opposing positions may differentiate themselves from each other, leading to political polarization. This combination of group polarization (within groups) and political polarization (within groups) provides a social psychological foundation for collective action and intergroup conflict. To prevent this, other more powerful groups may try to prevent communication.

**Fig. 3 | How supporters of two opposed positions polarize into two groups over time. a** Time 1: Prior to interacting; people who support two opposed positions on an issue. People who support one position are represented by the red dots, and people who support the other position are represented by the blue dots. **b** Time 2: People start to interact with other people who share their position on the issue, and form connections. **c** Time 3: Group polarization (within groups) and political polarization (between groups) start to occur. **d** Time 4: The groups expand as more people interact and connect with like-minded others, and the groups move further apart from each other. At the societal level, this results in political polarization.

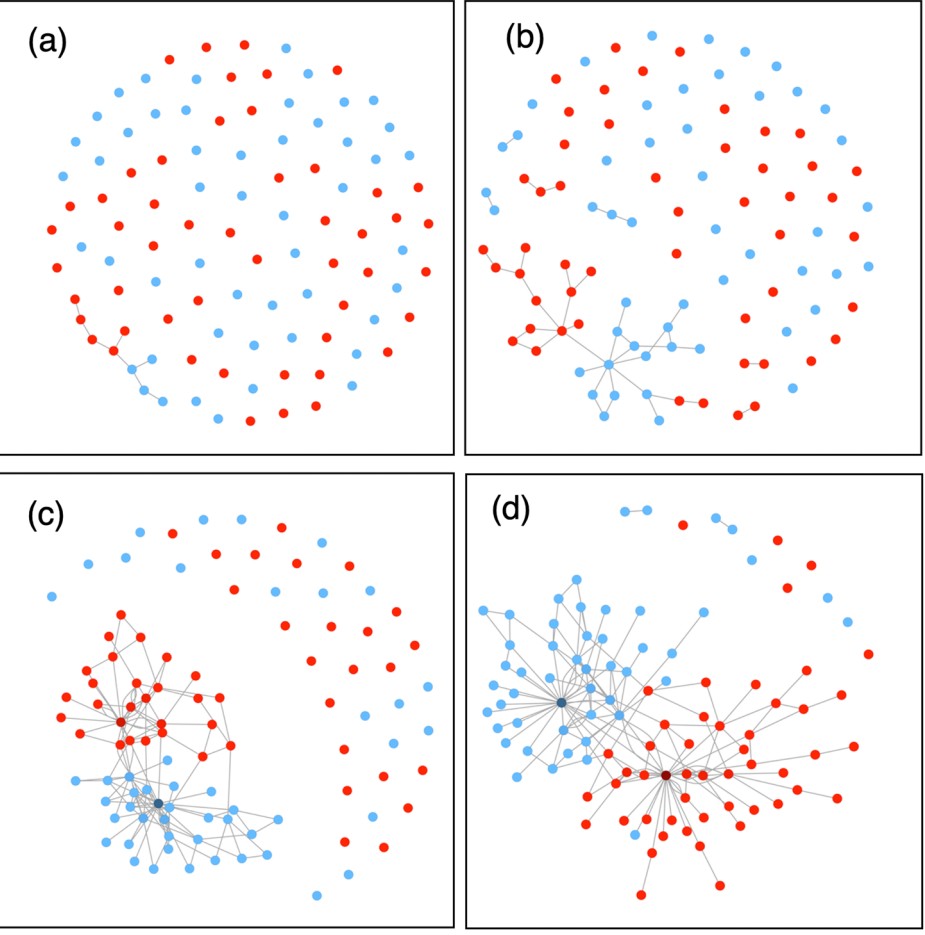

vehicle for myriad social and psychological variables that facilitate the processes and outcomes we describe below. For people who engage in collective action alone—but take collective action because it is on behalf of their group[59]—communication may take place in the form of reading or hearing and then internalizing the arguments and opinions of likeminded others. This is why some repressive regimes cut off access to SNSs to prevent rebellion; and some regimes implement offline curfews to prevent people meeting and communicating in person (Proposition 5).

Notwithstanding, the affordances of SNSs can amplify and facilitate the processes of communication underlying polarization. Before the internet and SNSs existed, communication was limited by reach, speed and scale, but the processes in Box 2 were substantively the same, and communication still facilitated collective action[60,61]. For example, during the Protestant Reformation, Luther and his supporters leveraged then-new new technologies (the printing press) to communicate their ideas. Therefore, communication offline and via SNSs are similar, but SNSs include affordances that amplify the processes of offline communication (sometimes) allowing for anonymity, increasing reach, allowing many people to talk simultaneously, faster, and on a much larger scale; access to information is more instant, and recommendation algorithms make encountering likeminded others and material more likely[62].

It is also the case that people do not communicate via SNSs in isolation from their other online and offline activities—people can cross-post on multiple networked applications, sites, and forums, as well as simultaneously communicating in multiple offline modes. This occurs both prior to and during collective action[38,63]. According to media multiplexity theory, the number of communication modes people use is positively related to the strength of the relationship between them and the people with whom they are communicating[64], suggesting that using SNSs

to communicate alongside other modes of communication may strengthen an ingroup, better equipping them to act collectively. Concurrently, offline events and communications shape and influence SNS interactions, and vice versa.

## Social interaction leads to the formation of new, opinion-based groups

While group polarization can occur through communication between members of pre-existing groups and shape attitudinal extremity within and between longstanding groups such as Democrats and Republicans[40,65] (termed by Postmes et al. as deductive influence[66]), a distinctive aspect of the current argument is that we suggest that such processes are also the engine room for the emergence of new, contextually relevant groups and identities (via what Postmes et al. term inductive influence). Opinions are, of course, often linked to other, pre-existing, broader and/or longstanding identities but that they are not reducible to them[57,67]. New groups and new identities enter the social context all of the time, for example, new groups formed to oppose vaccination and mandates to curve the spread of a novel coronavirus[68]. A key tenet of the social identity perspective is that people will use the available and contextually relevant information to make sense of a given social reality[69]. In the online environment, opinions are frequently the only clues to category membership[70]. Accordingly, we propose that new groups can emerge from interactions between people who share ideological opinions about an aspect of the social or political status quo[37]. Such groups can be characterized as opinion-based groups[2,36,71], comprised of people who are united in their opinion about a desired state of affairs (pro, support; anti, opposed)[72,73].

The group polarization experiments[13] showed that when people who share an initial common opinion or attitudinal stance discuss an issue, they

become more extreme together[74]. This extremization is partly because of the social frame of reference that exists within the group[34]: what it means to be a group member in that context (i.e., the identity content, or norm), is to express increasingly stronger opinions[58]. We also contend that it results in people being more staunchly committed to the group itself[58,70,74,75]. The prejudiced school children who participated in Myer and Bishop's group polarization research[15] likely did not just emerge from their interaction with more strongly racist attitudes—they would also have felt a greater sense of belonging and commitment to those other participants who shared their position. Polarization is the process, and group formation along opinion-based lines is an outcome[37,70]. Opinion-based groups are new, shared social identities, that are created through social interaction with likeminded others[76].

This brings us to the third proposition in Box 2. When people discuss and interact with like-minded people about changes that they want to see in the world, it allows them to reach consensus on emotions (e.g., anger), beliefs (e.g., on collective efficacy)[76], and actions[77,78], and those elements become an internalized aspect of a shared social identity[79–81] (Proposition 3). It is through communication that people can identify with and connect with others, reach consensus, and form (or affirm) shared identities based on those discussions (Proposition 1). Figure 3 depicts a parallel process of group formation and polarization for those who are supportive of a particular change or stance and those who are opposed to a particular change or stance.

## Opinion-based groups actively differentiate themselves from each other

Crucially, because new opinion-based groups are premised upon specific socio-political opinions and the need for social change, they contrast themselves with salient outgroups who disagree with that position, and are committed to seeing their group win[50] (Proposition 4). That is, proponents —on both sides—will seek to accentuate the similarities that exist between them and other ingroup members, and exaggerate the differences that exist with outgroup members[82]. Indeed, users may strategically use platform affordances of SNSs to marginalize or otherwise distance themselves from people with whom they disagree[61]. The social identity approach emphasizes that social categories are not just about who one stands with (one's ingroup) but also who one stands against (the contextually salient outgroup)[32], but also that these are constantly being updated as new information is used to refine, re-evaluate and/or strengthen our understanding and affiliations. Winning a democratic contest involves creating a social movement behind a candidate, party, or proposition. The winner needs to be seen, by enough supporters, to express a shared vision for the nation that is believed to be better than that which is expressed by alternative groups[83]. Such social movements are founded on opinions that are polarized through social interaction and debate—both with people who share the views (Fig. 1b) and against those who do not (Fig. 1d)[84].

Although pre-existing political categories such as Democrat and Republican can become intensified and repurposed, dramatic changes occur when there is division into new, competing groups[36] that have formed and polarized through social interaction[78]. Thus, group polarization that creates opinion-based groups, who understand and articulate their own position (primarily) in relation to that of an opposing group, creates political polarization (Proposition 4). This is a process of active differentiation, which involves communicating who the group is and what they stand for, and how they are different from other groups[34,35]. These basic social psychological processes can undermine social cohesion[32] and create psychological barriers to intellectual humility, open-mindedness, and cooperation with people on opposing sides[65,85].

## Opinion-based groups propel social change

The implications of the above arguments are that intragroup discussion can then lead to polarized views with political consequences, such as collective action, but also conflict with those with whom we disagree (intergroup conflict, violence)[77,86–89] (Proposition 6). Indeed, our research suggests that

recent rapid global changes have involved creating new social identities based on shared opinions about how the world should be[22,37,60,78,81,89]. For example, Thomas et al. showed that, when non-radical students were invited to a small-group discussion to plan ways to stop caged-chicken farms, the group discussion boosted the participants' shared view that caged farming was wrong and an increased proportion of them took political action by signing a letter addressed to the government[86]. Moreover, when groups were encouraged to consider that achieving change sometimes means breaking the law (involving active conflict with another group or authority), participants in the group discussion were more willing to consider unlawful action (compared to isolated individuals)[86]. Thus, a short, small group discussion can produce meaningful changes, but these shifts are most pronounced where group members also discuss the competing positions of opposing groups[88] (as in Fig. 1c, d). Bliuc et al.'s analysis of the debate between climate change skeptics and believers as two groups in conflict[2] is another good illustration of Doise's observation that competing groups can drive each other further apart[20]. Doise's findings are extended in Simon and Klandermans's argument that a conceptual triad is at the core of action for social change[84]. Power struggles involve an awareness of shared grievances for which the other side is blamed; and the two sides are involved in a contest for support from a third party (e.g., the general public, third parties or government)[84].

Given this background, we demonstrate in the remainder of the paper the power of new, polarized groups to produce social change in relation to three significant global examples: waves of mass protest movements; political violence; and right-wing populist movements. We illustrate how these recent, profound societal changes represent divisions into competing groups (Proposition 4). These groups were based on views that the status quo (and any authority that enforces it) was illegitimate and must be overthrown (Proposition 3). They were polarized through online channels (Proposition 1), and opposing groups attempted to silence them to disrupt their ability to take action (Proposition 5). In each case, collective action was the result (Proposition 6). The nature of the groups that emerged, and their norms for action, depended on their perceptions of the socio-political context and the nature of the ideas that they discussed[90]. We acknowledge that many other factors influence social and political changes, not least the qualities of the socio-structural context, but our focus here is on how and why social interaction can allow group and political polarization, and (therefore) mobilization to collective action.

## Connecting polarization and protest

Our arguments about the links between polarization and protest become clearer in the context of the history of the last ~15 years. In late 2010, a local Tunisian protest rapidly spread leading to regime change in Tunisia, Egypt and Libya, and civil war in Syria[91]. Images of unprecedented anti-government street protests were taken from camera phones to video sharing SNSs (YouTube) and then back across the Arab world through satellite television[92]. These movements achieved change by using technologies that showed dissent as normal, allowed activists to plan action on the street, and promoted revolutionary versions of national identity[60]. In late 2010, Tunisian activists were expressing their version of how they wanted the world to be in video and rap music[60] (much as 16th Century Lutherans challenged the Catholic Church with printed pamphlets and folk music[93]). The Tunisian revolution encapsulates our six propositions (Box 2).

The rapid, massive growth of the anti-regime movement was possible because online technologies provided people with a platform for group discussion[60]. Social media have both informational uses (people can share information to coordinate and organize events) but also play an important role in the social affirmation of opinion[38,94,95]. This platform enabled people to share their dissent (creating the conditions for the polarization of opinion) and to organize protests[60]. At the time, new technologies also disrupted the mechanisms for repressing (anti-regime) free speech and (protest) assembly. The Egyptian government could seek to disconnect the country from the internet, but it could not stop protesters from organizing

protests in Tahrir Square by SMS[96]. These channels allowed group polarization to occur despite the regime's efforts to the contrary. Repression works by preventing people from interacting with each other[96] (Proposition 1 linking to Proposition 5), precisely because it prevents the polarization of (anti-regime) opinion[60].

Since the Arab Spring revolutions, these methods and means have been replicated in waves of popular protest. Mass protests increased annually on a global basis between 2009 and 2019 across all regions all over the world[12,97]. Below, we briefly consider the evidence for the role of communication by online technologies in shaping the emergence of protest in three case studies: Kony2012, Occupy Wall Street, and the global solidarity movement to support Syrian refugees. Although these are not current examples or even the biggest protest events of the past decade, each were record-breaking at the time that they occurred; each illustrative of a broader set of processes that have happened on greater scale since; and each illuminates the six key propositions in Box 2, as we explain below.

In 2012, a video about the heinous activities of Ugandan Joseph Kony became the most viral (popularly shared) video of all time (at that time)[98]. Joseph Kony is suspected of 36 counts of war crimes and crimes against humanity, allegedly committed between at least 1 July 2002 until 31 December 2005 in northern Uganda[99]. The video was disseminated through YouTube and invited adherents to both share the video online through their networks (Facebook, Twitter), as well as prepare to attend an offline event (Cover the Night). In spite of the achievements of the online campaign, the offline event was not successful and the Kony2012 campaign is often now derided as the archetype of slacktivism (that is, low-cost online engagement that does not reflect sincere or enduring commitment[100]). Nevertheless, the same factors that predict engagement in traditional (primarily offline) forms of action also predicted engagement in the online and offline Kony2012 campaign[101]. That is, proponents were outraged by Kony's activities and his continued freedom; believed in the effectiveness of collective, coordinated actions; and were committed to a group based on a novel (new) yet shared, Anti-Kony opinion. Participants were driven to participate as an expression of collective selfhood. Furthermore, this commitment did not spring from a vacuum. It emerged from the reality of online interactions: through viewing, watching, discussing and sharing the online video, participants saw themselves as active agents in the battle to bring Kony to justice[101].

Since Kony2012, there are innumerable examples of people seeking to become the change that they sought in the world via acting with other people, from across the world, who share their views. Similar though unique patterns can be identified in the Occupy Wall Street movement[102], and the Ukrainian Euromaidan[103]. Protest about economic inequality and austerity is globally one of the most prominent forms of action[97]. Smith et al.'s analysis of the 5343 Facebook posts about Occupy show that these interactions were marked by heated interactions about the desired changes (an injunctive norm: revoke corporate personhood) and the prescribed actions necessary to bring that change about (Occupy wall street)[102]. To the extent that these positions were agreed upon and validated through online interactions, the interactions yielded qualitatively new groups defined along opinion-based fault lines: support or oppose Occupy (Box 2; Fig. 2).

Finally, similar patterns of rapid, global, social mobilization were witnessed in September 2015, when an image of a drowned Syrian child was disseminated globally through the traditional and social media. The image galvanized an outpouring of popular support and was implicated in dramatic policy reversals in many countries including Australia, the US, Canada, and the UK[104]. Most pertinently for current purposes, the patterns of online interaction appear to have played a driving role in this groundswell of support. Online engagement with the image and plight of Syrian refugees predicted the emergent (pro-refugee) group across six very different national contexts[105]. Smith et al. developed a longitudinal paradigm to analyse the Tweets of users before the emergence of the images of Aylan Kurdi, the week they emerged, and 10 weeks afterwards. The results showed that online interactions about the unacceptable harms and threat experienced by refugees sustained expressions of solidarity with refugees 10 weeks later[23]. The results affirmed the view that communicating online can promote sustained forms of psychological engagement. Nevertheless, it is also clear that, when these interactions stop, so too does the wave of popular support dissipate. Thomas, Cary, Smith, Spears and McGarty sampled people at the peak of the response to the image of Aylan Kurdi, and then one year later. They showed that *changes in* social media engagement explained reductions in commitment to the opinion-based group which, in turn, explained the reduced commitment to act[22].

In each of these cases (Kony2012, Occupy, Syrian refugee crisis), there have been rapid social changes based on sharing opinions about desired changes in the world[106]. Homogeneous clusters of opinion can rapidly grow on SNSs such as Facebook and create echo chambers where similar views are shared and reinforced and hence polarize[52] (as per Fig. 3). Online engagement helps opinion-based groups become ready for action through polarization and part of that readiness comes from linking up otherwise isolated local clusters into an ideologically coherent whole[74]. In effect, the internet allows political operators to run their own massive group polarization experiments by enabling people to share and discuss opinions, thus allowing new social movements to form rapidly.

The positive effects of these events have been overlooked in discussions of polarization. Entire countries were able to challenge authoritarianism and the events of the Arab Spring sparked speculation that new technologies would herald in a new global era of democratization[96,107]. In spite of their mixed legacy, Kony2012, Occupy and the waves of protest that followed showed how new technologies could be used to bring people together to agitate for changes to promote the rights and access of structurally disadvantaged people across the world. Yet, at the same time, it started to become clear that online technologies and interaction were not going to be the social panacea that was initially hoped for. Rather, as we discuss below, evidence emerged that the same tools that enabled engagement could be used to foster commitment to extreme forms of political violence. And, alongside efforts to use online technologies to promote greater rights and freedom for disadvantaged groups, people could also use them to promote divisive agendas around populism and hate. In the remainder of the paper, we focus on connecting our six propositions about the underlying nature of group and political polarization to explain how similar processes that enable global solidarity and social change, can also inspire political violence and anti-democratic sentiment, and hate.

## (Online) Polarization and political violence

Much of the public commentary about polarization on SNSs suggests putative links with political violence, including the sensational storming of the US Capitol on January 6, 2021[108,109]. Indeed, there has been an upsurge in political violence in the last decade[110], and a key factor behind this is the innovative way in which far-right groups such as QAnon[111], Proud Boys[112] and Stormfront[113], and groups such as Al Qaeda and Islamic State (IS)[114–117], have outsourced the dissemination of their propaganda to their decentralized networks of online supporters. For example, IS's online community are known as the media mujāhidīn[118]. Communicating via SNSs such as Telegram, the media mujāhidīn have broadcasted extreme positions and propaganda, clustered in aggregates[119], and discussed their views in encrypted chatrooms[114,115,120]. This enabled these violent extremist groups to recruit new adherents and inspire attacks, for example, the perpetrator of the 2017 Manchester terrorist attack, who killed 22 people by detonating an improvised explosive device during a concert in Manchester Arena, UK[121]. The perpetrator was critically influenced by his peers and family members, whose electronic devices were found to contain a significant volume of violent extremist material from groups such as Islamic State[122]. The media mujāhidīn both take advantage of, and have been forged by, SNSs' opportunities to share and validate extreme opinions (e.g., through re-posting and liking), with limited capacity for authorities to intervene[116,117]. Not surprisingly, given the results obtained by Thomas et al.[86], polarization of opinion within sympathetic communities has been the result. Similarly, through normalizing racist discourse and othering of non-White populations, Stormfront changed from being just a website on which White supremacists communicated racist views, to being a meaningful psychological group[113]

that was premised upon communicating their hatred of non-White out-group members. QAnon have also successfully used social media interactions to polarize and mobilize individuals around conspiracy theories, particularly on Parler[123,124], creating an online milieu that captured the imagination of people who were disillusioned with mainstream politics and explanations in the media[125].

To connect this example to our propositions in Box 2, there are three notable elements of the psychological environment within which these groups operate online. First, some group members use social media to express opinions and desire for intergroup actions that cannot be expressed in mainstream channels (e.g., violence), and those types of online interactions are related to the formation of new, polarized and mobilized psychological groups[21,126,127] (Proposition 1) and political activism[23,128] (Proposition 6). This is supported by laboratory research that shows that social interaction about outgroups increases action confidence and intergroup discrimination[87]. Indeed, engagement in social media discussions can transform political participation through transforming perceptions of personal and collective political efficacy[129].

Second, this connection to opinion-based groups through social media means that while radicalized internet users may be physically isolated, they are not psychologically isolated (as per Propositions 1–2)[21]. When people are visually anonymous online and a social identity is salient, they are likely to communicate with others along group lines[130,131]. This means that even when an individual engages in online chat alone, they can be radicalized through group conversations and group polarization processes.

Third, and overall, the global increase in political violence[110] may be fuelled, in part, by a process of group polarization resulting in political polarization (or co-radicalization[132]) (Proposition 4). For example, anti-Muslim views espoused by White supremacists and some political leaders are exploited by groups such as IS in their online propaganda to promote more extreme views[118]. Conversely, far-right leaders use IS actions in their own propaganda to promote anti-Muslim attitudes. In 2017, Marine Le Pen, leader of the French far-right group National Front, used her Twitter posts to attempt to increase perceptions of a "threatening ethnoreligious other" and in turn bolster support for her party (p. 131)[133]. This use of outgroup actions to promote ingroup polarization works in the same way as attacking climate change sceptics would be expected to solidify opposition to climate change action[2]: it actively differentiates the ingroup from the outgroup (Proposition 4), and provides the rationale for collective action (Proposition 6).

As populism, including anti-Muslim sentiment, increased in Western democracies, the IS propaganda narrative online shifted towards framing themselves as embattled, but defiant[134]. IS used evidence of discrimination of Muslims in the West to recruit new supporters. For example, in their propaganda IS highlighted Western discrimination against Muslim women, such as forced unveiling practices, to recruit Western women as foreign fighters (the muhajirat)[135]. Loken and Zelenz explored the cases of 17 muhajirat, most of whom were radicalized on SNSs such as Twitter, Tumblr, and Kik through interacting with IS members about their feelings of isolation and experiences of discrimination in their Western states[135]. IS then used the *muhajirat* they had successfully recruited as positive examples of women who had chosen a life in the Caliphate to illegitimize the Western narrative that IS mistreat women and girls. Thus, like with White supremacists, it was communication about the intergroup context that led to polarization of opinion within the group (i.e., group polarization) and hardened one group further against the other (i.e., co-radicalization): the process of group polarization resulted in political polarization.

## Polarization and populism: reactionary hate
Large-scale polarization can also be seen in the successes of right-wing populist (RWP) movements in Western democracies[136,137]. These movements used similar social media strategies to those seen during the Arab Spring[138]. We propose that online polarization of opinion helped RWP movements grow in two main ways. First, RWP leaders reframed divisive

and discriminatory rhetoric as free speech. That is, positions previously suppressed by laws and custom were shared and discussed, so that they became justified and normalized[139], group polarization being therefore enabled (in line with Proposition 3). SNSs (especially Facebook) provided platforms for millions of posts expressing positions that could not be readily expressed in mainstream media, including millions of fake news posts in the US in 2016[140], creating homogeneous clusters of opinion-based networks (or echo chambers[52], Fig. 3). Prior to the 2016 election, it was estimated that US citizens were exposed to 38,000,000 items of fake political news on social media; most (78%) expressing pro-Trump/anti-Clinton views[141] (although fake news was unlikely to have shifted the election result[141,142]). This polarization, fostered through mass communication channels, created unity amongst those who shared opinions on the populist agenda (in line with Proposition 1).

Second, SNSs were used to connect supporters in regional areas without the need for expensive door-to-door campaigns on the ground. Through online channels, views that were previously isolated (and perhaps not told to opinion pollsters) spread and became reinforced by similar views held by millions of (possibly fake) others in the same local community. The clustering played out geographically in the US Presidential election in 2016[143]. Why was this geographical and opinion clustering so important? By tracking referrals from browsers to top fake news sites (that published stories such as "Clinton sold guns to Isis" and "Obama to ban pledges of allegiance in schools"), Fourney et al. found a very high referral rate from Facebook and Twitter (68% of all page visits) and an extremely high correlation between the number of visits from the location of the referring browsers (Internet Explorer and Edge browsers that were available to the authors who were Microsoft employees) and the proportion of voters voting for the Republican candidate ($r = 0.85$ at the state and county level)[143]. In other words, there were geographic clustering of patterns of online behaviour in the five months before the presidential election, and this affected political behaviour (voting). Voters in states that were won by Republicans were much, much more likely to navigate from Facebook and Twitter to fake news websites. Polarization, catalysed by online communications, thus led to the mobilization of political action.

Simulations of networks and empirical results provide some guidance on how this can play out. There is evidence that echo chambers exist on social media and play a crucial role in polarization[52,56], and work by Törnberg provides additional clarity. Using simulated networks and modelling of empirical data on Twitter (retweets of messages by politicians in 37 countries), Törnberg showed the synergetic effects of opinion polarization and echo chambers on the virality of messages[144]. Specifically, misinformation was more likely to spread when biased (polarized) clusters encountered information in homogeneous environments. This can be interpreted as a failure for error correction and contestation of extreme views in closed communities, enabling a perception of unity around populist (and other) opinions (Proposition 1).

Therefore, it is likely that RWP leaders became popular by exploiting current divisions, uniting people around their concerns, and using social interaction to create a polarized social movement. Perhaps most importantly, they brought previously hidden views into the open. Polarization is not inevitable: uncommon views need not become popular, but in line with Proposition 3, it is hard for those views to grow in popularity without communication[87]. Positions that are not communicated do not polarize.

## Outlook
Our analysis suggests that group polarization can lead to political polarization through a process of discussion about ways to change the world that unites individuals around a common cause, and actively differentiates them from other groups. This process provides the psychological foundation for mobilization to collective action. A corollary of our arguments is that polarization and collective action can be progressive (promoting greater rights, access, freedoms for people), promoting revolutionary change in ways that are good for democracy; and reactionary (seeking to protect rights and access of privileged people and groups), promoting oppressive change

in ways that may see societies slide back towards authoritarianism, or encourage commitment to political violence. This process need not favour or encourage any particular political views: indeed, our examples show that the same processes have occurred in a variety of different groups and contexts. Rather, it is the unity of beliefs within the group, and the social psychological transformation of them through discussion, that matter. Therefore, polarization should not be pathologized as a social ill[145], like how Le Bon once pathologized crowd behaviour as irrational[9], or how Janis pathologized group decision making as dysfunctional[146]. Just as groups can make good decisions and people within crowds can act sensibly and rationally, polarization is not bad, in and of itself, and can be a tool for positive social change. Indeed, a society with no division and polarization, where citizens are homogenous in their views, would equate with a static, totalitarian society with no prospect for change. Nevertheless, many people are wondering how ideas that they thought had been consigned to the past have become so popular again. Why, for example, are racist, sexist, pro-violence, and anti-science views returning in industrialized societies? Is it because we live in an era where certain ideas are just more appealing to the population, or because certain political leaders are craftier than others? Perhaps, but as we have sought to illustrate, some of the reasons for current events rest on processes that were brought to light by psychological science decades ago, but now play out through new technologies. We do not believe those technologies work better for some ideas than others. If Victor Hugo was right that "there is nothing so powerful as an idea whose time has come", then those times are most likely to come—for both bad and good ideas—when opportunities to share and reinforce them abound.

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

## Author contributions

This statement of author contributions was prepared using the CRediT (Contributor Roles Taxonomy) guidelines. Laura Smith, Emma Thomas, and Ana-Maria Bliuc contributed equally to: Conceptualization, writing—original draft, visualization, and are therefore joint first authors. Craig McGarty contributed to: Conceptualization, writing—original draft, visualization.

## Competing interests

The authors declare no competing interests.
