## [Peer Review File · Communications Psychology]

8th Nov 23

Dear Professor Smith,

Thank you for your patience during the peer-review process. Your manuscript titled "Polarization is the Psychological Foundation of Collective Engagement and Extremism" has now been seen by 3 reviewers, and I include their comments at the end of this message.

The reviewers are in principle enthusiastic about your work. However, they also mention a number of concerns. We are very interested in the possibility of publishing your manuscript in *Communications Psychology*, but would like to consider your response to these concerns in the form of a revised manuscript before we make a decision on publication.

In detail, we ask you to address all points raised by the reviewers. We highlight in particular the need to ensure that processes/groups are presented as not inherently positive/negative where there is room for debate on the matter (e.g., in relation to the process of collective action and the use of the label "extremists"); moreover, a review-type article needs to ensure timeliness by incorporating the newest relevant literature on a topic. Reviewer #1 makes a comment to that end, and we ask you to carefully review the cited literature to ensure that novel contributions that inform your framework are suitably recognized. We also strongly recommend making use of more Figures - Reviewer #2 commented on this issue.

Finally, to ensure that we can process your revisions efficiently we request that you ensure that it complies with our formatting requirements, which you will find following this link:
<https://www.nature.com/documents/commsj-style-formatting-checklist-review-perspective.pdf>

* **TRANSPARENT PEER REVIEW:** *Communications Psychology* uses a transparent peer review system. This means that we publish the editorial decision letters including Reviewers' comments to the authors and the author rebuttal letters online as a supplementary peer review file. We publish these records for all accepted manuscripts. However, on author request, confidential information and data can be removed from the published reviewer reports and rebuttal letters prior to publication. If your manuscript has been previously reviewed at another journal, those Reviewers' comments would not form part of the published peer review file.

If you have any questions about any of our policies or formatting, please don't hesitate to contact me.

Please use the following link to submit your revised manuscript and a point-by-point response to the referees' comments (which should be in a separate document to any cover letter):
[Link redacted]

We hope to receive your revised paper within 12 weeks; please let us know if you aren't able to submit it within this time so that we can discuss how best to proceed. If we don't hear from you, and the revision process takes significantly longer, we may close your file. Of course, if you are unable to estimate, we are happy to accommodate necessary extensions nevertheless.

Please do not hesitate to contact me if you have any questions or would like to discuss these revisions

further. We look forward to seeing the revised manuscript and thank you for the opportunity to review your work.

Best regards,

Antonia Eisenkoeck
Senior Editor
Communications Psychology

REVIEWERS' EXPERTISE:

Reviewer #1: polarization
Reviewer #2: polarization
Reviewer #3: collective action

REVIEWERS' COMMENTS:

Reviewer #1 (Remarks to the Author):

This paper presents a novel and very interesting new perspective on the nature of political polarization, which it argues can be understood as a function of group polarization. The paper is clear and well-written and draws on an extensive body of empirical work to justify the arguments made. I think readers in social psychology and other relevant social sciences would be interested in this paper. The history of group polarization research alone is fascinating and should be widely read. The argument that political polarization itself should not be equated with social dysfunction/ reaction was a refreshing one (it reminded me of the point made by Aurelien Mondon that the problem is not 'polarization' but right-wing extremism).

I had a few questions/ suggestions that the authors might want to consider in a minor revision.

p. 13 'we propose that group polarization is more than simply extremization of prior or pre-existing groups: it represents the creation of a new or transformed social psychological entity that is premised upon consensus on the ingroup (polarized) position'.

I found this argument a bit confusing. A premise of the paper is that political polarization is based on the processes we know as group polarization. After arguing this, the authors seem to be amending the scope/ definition of group polarization. Aren't many the groups in the group polarization literature pre-existing groups/ categories?

In a few places, I thought the arguments could be further/ better justified by reference to the relevant social identity literature. For example, I wondered why the proposition 4 discussion (p 13) didn't say more about basic social identity and self-cat processes (accentuation, prototypicality etc.) with the relevant references, as this is what is being referred to.

Similarly: p. 21 'Online radicalization towards support for political violence is thus a product of group polarization, even when an individual engages in online chat "alone".'

Again this kind of claim could be justified by citing some of the earlier SCT work (for example Postmas and Spears on computer mediated communication).

A related but bigger point is whether the process of group polarization in that literature is really the same as the process identified in the collective action literature cited in the paper. EMSICA and the research following it which the authors cite is about identity content rather than extremity, isn't it? It's notable that in explaining the power of interaction, the authors don't really refer to the SCT explanation of group polarization.

Minor points

p. 5 'now neglected¹⁵' is a reference from 22 years ago. A more recent reference might better illustrate that group polarization is currently neglected (maybe a textbook?).

p. 10 'The creation of new opinion-based groups via group polarization (as per #2), who articulate their own position (only) in terms of how it differs to that of an opposing group; leads to political polarization'

Check punctuation.

p. 20 'lone wolf attacks'

I understand that that this term is no longer favoured by the relevant experts and researchers. Its assumptions also seems in conflict with the group-based approach of the current paper.

pp. 20-21 I wonder about the term 'extremism'. The UK government and others seems to use it about activists the authors might not (such as environmental campaigners). Or, turning it around, if the authors are right that group polarization underpins political polarization then all groups discussed in the article are 'extremists' (is that ok? It sounds a bit problematic when many such groups are trying to avoid being marginalized – for example by placing debates around the climate crisis in the mainstream).

p. 22 '(i.e., akin to group polarization'

Why 'akin'? I thought the argument was it just is group polarization.

John Drury

Reviewer #2 (Remarks to the Author):

I had the pleasure of reading the manuscript "Polarization is the Psychological Foundation of Collective Engagement and Extremism". In this article, the authors describe how group polarization can give way to political polarization, which ultimately might result in collective action. They specifically focus on the role of social media in this process. To make their claims, they draw both on literature and real-world examples from across the globe. Below I outline some recommendations for how to strengthen the theoretical arguments proposed.

1. The authors list six propositions for the connection between group think, polarization, and collective action (see table 2). Overall, I think that these propositions make a lot of sense and are clearly thought out. However, I would appreciate more acknowledgement of the interconnectedness between them. For instance, based on the way it is currently written, it sounds like these are six steps that occur in the order listed; however, many of the early steps serve as precursors for the later steps. For instance, as noted in the table, proposition 6 can be an outcome for proposition 1, 2 and 4. I recommend that rather than listing the propositions as a table, that the authors convert it to a diagram that highlights pathways between each proposition. I also recommend explaining such a figure in the text as well.

2. I am curious about the relationship between group polarization, political polarization, and collective action in contexts that have limited or no access to social media. The authors allude to this possibility when they discuss how Egypt cut off access to internet (see page 16), however, in this context, social media did still play an important role in propelling initial calls to collective action. Do similar processes/links between group polarization, political polarization and collective action exist in absence of SNS, or is it something unique about social media that strengthens these connections? I am not necessarily recommending that the authors spend a lot of time on this in the paper, but I find it important to discuss. Relatedly, SNS do not exist in isolation.

3. I appreciate how the authors list the different definitions of polarization and clarify that they refer to political polarization as "opinion-based" (see page 9). However, the other definitions of polarization are

also particularly relevant to the arguments made. For instance, typically opinion-based/issue-based polarization can give way to affective polarization. Likewise, collective action can be fueled by co-radicalization, which is likely also fueled by group polarization. Focusing on one type of polarization has the potential to limit theoretical impact of the current manuscript. I recommend either providing stronger justification for why issue-based polarization is the specific polarization of focus or integrating the other types of polarization into the model.

Reviewer #3 (Remarks to the Author):

Thanks for the opportunity to review this work. I really enjoyed reading this manuscript and found a lot to appreciate about it. The paper is on a very interesting and timely topic, and the integration of both the political polarization and group polarization literatures is a welcome addition as these are often situated in different areas when there are valuable synergies between these areas of study. I also found the paper very nicely written keeping the reader engaged throughout. I only have two comments about the paper in its current form that the authors should be able to address in a revision.

First, while I really like the idea to present the work as taking a balanced approach to examining the effects of polarization, there are aspects of the current framing that may be too simplistically valenced. For example, in the early introduction, the promotion of collective action is framed as if it is inherently good, while collective action can lead to positive outcomes or disastrous ones. Even movements that may have seemingly good ideals and goals can turn toxic and destructive through polarization. For example, despite promises of greater collective welfare and giving power to peasants, polarization emerging from social movements in China and Cambodia led to death of millions. While seemingly extreme, the point is that social change emerging from collective action is not inherently good or bad. Anti-vax protests during the pandemic may have perceived their actions as positive for standing up against what they deemed to an authoritarian state, but it may have simultaneously increased risk of disease among the wider public and been destructive to others. Therefore, it would be better to further nuance the writing to clarify that polarization has the potential to bring people together to engage in collective action, but it is agnostic on whether such collective action can end up in destruction or human progress. I would argue that while I see what the authors are getting at here about its potential benefits, there is a risk that polarization undermines social cohesion and cuts people off from each other making it difficult to perspective take and understand each other in a liberal democracy with a pluralistic populace. However, I understand what the authors are meaning to contribute to the literature and suggest nuancing the writing without an insinuation of the valenced nature of social change emerging from collective action.

On a second, more minor point, the framing of page 23 about RWP leaders attacking the consensus that criticizing minority members as racist is difficult to understand. What types of criticism of minority groups are racist? Any criticism? While there has been a concept creep (see Haslam et al., 2020; Haslam, 2016) over the meaning of racist, this seems like an odd choice of words. I write this review in the midst of a global conflict that has led to a spike in antisemitism worldwide, and some argue that almost any criticism of Israel as antisemitic as the sole Jewish state in the world, while others see it as entirely justified, so who gets to decide what constitutes free speech and what is hate speech? The larger point is that what is deemed as bigotry is in the eyes of the beholder.

Besides these relatively fixable issues, I thought the paper read well and would make a useful contribution to the literature. I hope this work will encourage more empirical research on this useful topic of study.

Response to Reviewers

COMMSPSYCHOL-23-0296, "Polarization is the Psychological Foundation of Collective Engagement"

Below, we outline the reviewers' suggestions (indented, in italics) and how we have addressed them in this revision (in bold).

Reviewer #1:

This paper presents a novel and very interesting new perspective on the nature of political polarization, which it argues can be understood as a function of group polarization. The paper is clear and well-written and draws on an extensive body of empirical work to justify the arguments made. I think readers in social psychology and other relevant social sciences would be interested in this paper. The history of group polarization research alone is fascinating and should be widely read. The argument that political polarization itself should not be equated with social dysfunction/ reaction was a refreshing one (it reminded me of the point made by Aurelien Mondon that the problem is not 'polarization' but right-wing extremism).

Thank you for these positive words. We appreciate the tip about Aurelien Mondon's work, and we now cite his article in relation to the above point on p. 28.

I had a few questions/ suggestions that the authors might want to consider in a minor revision.

1. p. 13 'we propose that group polarization is more than simply extremization of prior or pre-existing groups: it represents the creation of a new or transformed social psychological entity that is premised upon consensus on the ingroup (polarized) position'. I found this argument a bit confusing. A premise of the paper is that political polarization is based on the processes we know as group polarization. After arguing this, the authors seem to be amending the scope/ definition of group polarization. Aren't many the groups in the group polarization literature pre-existing groups/ categories?

Thank you for raising this - it is an important point that we should clarify. The simplest way to explain this is through seeing group polarization as a process rather than an outcome, with a starting point (pre-existing opinions, social categories, or groups) and a potential end point of emergent / transformed groups. These new groups can't be understood by referring (only) to the original ("starting point") group or static social categories.

The advantage of conceptualizing group polarization as a process is one can separate the polarized attitudes from the other social psychological outcomes associated with the shared and concurrent experience of extremization, i.e., a feeling of becoming more extreme together, understanding each other's positions in more depth, sharing those positions, and bonding over them. These new social psychological phenomena that result from polarization cannot be understood only with reference to pre-discussion similarities/categories. Polarization is the *process*, identity formation can be an outcome.

We now cite evidence that demonstrates that the opinions shared during discussions often go over and above those that are associated with pre-existing category memberships. For example, on pp. 14-15 we explain:

“Whilst group polarization can occur through communication between members of pre-existing groups and shape attitudinal extremity within and between longstanding groups (e.g., Democrat and Republican^{1,2} - what Postmes et al.^{3,4} term “deductive influence”), a distinctive aspect of the current argument is that we suggest that such processes are also the engine room for the emergence of new, contextually relevant groups and identities (via what Postmes et al. term “inductive influence”). Opinions are, of course, often linked to other, pre-existing, broader and/or longstanding identities but they are not reducible to them^{5,6}. New groups and new identities enter the social context all of the time: in November 2019, few of us could have imagined global social movements forming around a novel coronavirus. A key tenet of the social identity perspective is that people will use the available and contextually relevant information to make sense of a given social reality⁷. In the online environment, opinions are frequently the only clues to category membership⁸. Accordingly, we propose that new groups can emerge from interactions between people who share ideological opinions about an aspect of the social or political status quo⁹. Such groups can be characterized as *opinion-based groups*¹⁰⁻¹², comprised of people who are united in their opinion about a desired state of affairs (pro, support; anti, opposed)^{13,14}.”

Indeed, we may go so far as suggesting that many of the groups in the group polarization literature could be understood as opinion-based groups (being anti-Black, pro-supporting the war effort) but they were not recognized as such because the concept didn’t exist at that time:

“The group polarization experiments¹⁵ showed that when people who share an initial common opinion or attitudinal stance discuss an issue, they become more extreme together¹⁶. This extremization is partly because of the social frame of reference that exists within the group¹⁷: “what it means” to be a group member in that context (i.e., the identity content, or norm), is to express increasingly stronger opinions¹⁸. We also contend that it results in people being more staunchly committed to the group itself^{8,16,18,19}. The prejudiced school children who participated in Myer and Bishop’s²⁰ seminal group polarization research likely did not just emerge from their interaction with more strongly racist attitudes - they would also have felt a greater sense of belonging and commitment to those other participants who shared their position. Polarization is the process, and group formation along opinion-based lines is an outcome^{8,9}. Opinion-based groups are new, shared social identities, that are created through social interaction with likeminded others²¹.” (p. 15)

2. In a few places, I thought the arguments could be further/ better justified by reference to the relevant social identity literature. For example, I wondered why the proposition 4 discussion (p 13) didn’t say more about basic social identity and self-cat processes (accentuation, prototypicality etc.) with the relevant references, as this is what is being referred to.

We now include more references to the social identity literature to justify the arguments. In the primary sections where we articulate our propositions, we have now clarified how this approach draws upon seminal theorizing from within the social identity tradition. For example, on page 13 we state that:

“These observations are consistent with the social identity approach – that the groups that we stand “with” (versus those whom we are “against”) are those with whom we feel we belong, but also those groups that help us to best make sense of the social context that confronts us^{18,22}.”

In the section on, “Social interaction leads to the formation of opinion-based groups” (pp. 14-15), we clarify that a key aspect of the social identity approach is that people will use the available information to make sense of social reality but that, online, opinions are the only available social information (and hence become a primary basis for social categorization).

“A key tenet of the social identity perspective is that people will use the available and contextually relevant information to make sense of a given social reality. In the online environment, opinions are frequently the only clues to category membership.”

Following the suggestion of Reviewer 1, we also now mention prototypicality and accentuation, although we have been conscious of keeping our explanations as accessible as possible for our more generalist audience, so we avoid complex terminology where we can. We now write:

“This extremization is partly because of the social frame of reference that exists within the group¹⁷: “what it means” to be a group member in that context (i.e., the identity content, or norm), is to express increasingly stronger opinions¹⁸. We also contend that it results in people being more staunchly committed to the group itself^{8,16,18,19}.” (p. 15)

In the section on, “Opinion-based groups actively differentiate themselves from each other”, we now clarify that one of the outcomes is accentuation:

“That is, proponents – on both sides – will seek to accentuate the similarities that exist between them and other ingroup members, and exaggerate the differences that exist with outgroup members²³.” (p. 16)

But also that these processes are not static or set in time but are constantly updating:

“The social identity approach emphasizes that social categories are not just about who one stands “with” (one’s ingroup) but also who one stands “against” (the contextually salient outgroup)²², but also that these are constantly being updated as new information is used to refine, re-evaluate and/or strengthen our understanding and affiliations.” (p. 16)

3. Similarly: p. 21 ‘Online radicalization towards support for political violence is thus a product of group polarization, even when an individual engages in online chat “alone”.’ Again this kind of claim could be justified by citing some of the earlier SCT work (for example Postmas and Spears on computer mediated communication).

Thank you for this suggestion, we now cite SIDE research^{24,25} on p. 24:

“...this connection to opinion-based groups through social media means that whilst radicalized internet users may be physically isolated, they are not psychologically isolated (as per Propositions 1-2)¹⁷. When people are visually anonymous online and a social identity is salient, they are likely to communicate with others along group lines^{119,120}. This means that even when an individual engages in online chat “alone”, they can be radicalized through group conversations and group polarization processes.”

4. A related but bigger point is whether the process of group polarization in that literature is really the same as the process identified in the collective action literature cited in the paper. EMSICA and the research following it which the authors cite is about identity content rather than extremity, isn’t it? It’s notable that in explaining the power of interaction, the authors don’t really refer to the SCT explanation of group polarization.

Below, we explain how our conceptualization of the process of group polarization integrates and builds upon ideas from the SCT/RII literature, and from EMSICA. We have made the edits described in our response to Reviewer 1 #2, above, to explain how our approach is consistent with the SCT approach.

The SCT/RII explanation of polarization focused on attitude extremity (as an *outcome*). There was a focus on pre-existing, sometimes abstract, social categories (perhaps because SCT/RII predated Web 2.0 and the proliferation of research on opinion polarization and networks/echo chambers that followed). One of our novel arguments is that group polarization (the extremization *process*) leads to the development of new social psychological phenomena (pp. 14-16): novel identities, homogeneity, identifications, and identity content (including norms for action):

“This extremization is partly because of the social frame of reference that exists within the group¹⁷: “what it means” to be a group member in that context (i.e., the identity content, or norm), is to express increasingly stronger opinions¹⁸. We also contend that it results in people being more staunchly committed to the group itself^{8,16,18,19}.” (p. 15)

As Reviewer 1 points out, unlike SCT/RII, EMSICA^{26,27} (cited on p. 15) does not make arguments about extremization, per se. EMSICA explains how “identity content” (perceptions of efficacy and injustice) drives commitment to a group (identification). Therefore, EMSICA can be leveraged to explain how people become more committed to a group through discussion about relevant social issues. EMSICA implies that you can’t separate identity content - the meaning of the identity - from identification. We extrapolate from EMSICA’s ideas to propose that you cannot separate the process of polarizing from the outcomes of polarization (e.g., ingroup homogeneity, commitment to the group, etc).

We believe it is timely to combine these insights. The SCT/RII propositions pre-dated a huge expansion in polarization research and much new theoretical development (including the work of Postmes et al., EMSICA and opinion-based groups). On page 6, we explain that one of the motivations for this manuscript was that:

“The group polarization phenomenon was isolated just before a massive expansion in the study of group processes derived from the social identity perspective^{22,28}. More than fifty years after the paper by Moscovici and Zavalloni¹⁵, we now know much more about such identities, and how those group identities form and impact on attitudes and behaviors. Whilst important steps have been taken to link that knowledge with the concept of group polarization in the work of Postmes, Haslam and Swaab³ and others, the connection of the identity processes involved in group polarization^{17,29} to political polarization is, as yet, weakly articulated. As we will argue below, processes of group polarization are fundamental to group formation, group life, mobilization, and social change, and part of social change involves forming groups based on opinions¹² about changing the world⁹.”

Therefore, a major contribution of this *Perspectives* article is to integrate insights from group polarization research, EMSICA, and SCT/RII to understand the connections between group polarization, political polarization, and collective action.

Minor points

5. p. 5 ‘now neglected¹⁵’ is a reference from 22 years ago. A more recent reference might better illustrate that group polarization is currently neglected (maybe a textbook?).

We have now removed the phrase “now neglected” from the manuscript. We were not able to identify another more current source for this (and publication data do not have good coverage over this timeframe). This point was not a central part of our argument.

*6. p. 10 ‘The creation of new opinion-based groups via group polarization (as per #2), who articulate their own position (only) in terms of how it differs to that of an opposing group; leads to political polarization’
Check punctuation.*

We have amended this sentence as follows:

“The creation of new opinion-based groups via *group polarization* (as per #2) - who articulate their own position in terms of how it differs to that of an opposing group - leads to *political polarization*.” (Table 2, p. 10)

*7. p. 20 ‘lone wolf attacks’
I understand that that this term is no longer favoured by the relevant experts and researchers. Its assumptions also seems in conflict with the group-based approach of the current paper.*

We agree and have removed the term “lone wolf”.

8. pp. 20-21 I wonder about the term ‘extremism’. The UK government and others seems to use it about activists the authors might not (such as environmental campaigners). Or, turning it around, if the authors are right that group polarization underpins political polarization then all groups discussed in the article are ‘extremists’ (is that ok? It sounds a bit problematic when many such groups are trying to avoid being marginalized – for example by placing debates around the climate crisis in the mainstream).

We agree with this point, and as there is contention in both the academic literature and policy about definitions of “extremism”³⁰, we have removed the term “extremist” and “extremism” from the manuscript, including the title. We instead focus on attitudinal “extremization”, as per group polarization research, but we are very clear that this terminology is not to be conflated with violent extremism as an outcome:

“It’s important to note at the outset that attitudinal extremity (attitudinal extremism) is not the same as support for violent extremism, although the two are often conflated³⁰, a point we return to below.” (p. 4)

As Reviewer 3 suggested (#1), movements with seemingly prosocial ideals can engage in harmful forms of collective action. We agree, and on p. 25 of our original submission, we explained that:

“A corollary of our arguments is that polarization and collective action can be progressive (promoting greater rights, access, freedoms for people), promoting revolutionary change in ways that are good for democracy; and reactionary (seeking to protect rights and access of privileged people and groups), promoting oppressive change in ways that may see societies slide back towards authoritarianism, or encourage commitment to political violence. This process need not favour or encourage any particular political views: indeed, our examples show that the same processes have occurred in a variety of different groups and contexts.”

We are happy to have the opportunity to make this clearer in the revision. We have now included the following sentences in the manuscript:

“These basic social psychological processes can undermine social cohesion²² and create psychological barriers to intellectual humility, open-mindedness, and cooperation with people on opposing sides^{2,31}.” (p. 17)

“This “extremization”³² that occurs through group polarization indicates a shift towards a more extreme end of a continuum of beliefs, attitudes, or opinions, as demonstrated via the group polarization studies^{15,20}.” (p. 9)

Therefore, this does not place value upon the political orientation of the group. We note on p. 9 that, “Any group can undergo this process of extremization via polarization, but this does not mean that they are “extremists”³⁰, or that they support the use of violence. The nature of collective action that follows these processes depends upon the social context and the norms that form within those groups³³, and can vary from benevolent support, conventional activism, to political violence. Thus, such collective action can have a range of impacts, from progress to conflict and harm.” This point is also illustrated through our examples (p. 18 onwards).

*9. p. 22 ‘(i.e., akin to group polarization’
Why ‘akin’? I thought the argument was it just is group polarization.*

We agree and have deleted the words “akin to” from this sentence.

Reviewer #2:

I had the pleasure of reading the manuscript “Polarization is the Psychological Foundation of Collective Engagement and Extremism”. In this article, the authors describe how group polarization can give way to political polarization, which ultimately might result in collective action. They specifically focus on the role of social media in this process. To make their claims, they draw both on literature and real-world examples from across the globe. Below I outline some recommendations for how to strengthen the theoretical arguments proposed.

1. The authors list six propositions for the connection between group think, polarization, and collective action (see table 2). Overall, I think that these propositions make a lot of sense and are clearly thought out. However, I would appreciate more acknowledgement of the interconnectedness between them. For instance, based on the way it is currently written, it sounds like these are six steps that occur in the order listed; however, many of the early steps serve as precursors for the later steps. For instance, as noted in the table, proposition 6 can be an outcome for proposition 1, 2 and 4. I recommend that rather than listing the propositions as a table, that the authors convert it to a diagram that highlights pathways between each proposition. I also recommend explaining such a figure in the text as well.

In line with Reviewer 2’s suggestion, we now include a new figure (Figure 2), which visualizes the connection between the propositions in Table 2. We do not suggest that the propositions are stages or steps that occur in a specific order, but rather have tried to use the propositions to itemise, tease apart, and clarify the psychological processes that occur to connect the process of group polarization with collective action. We now clarify this on p. 9.

2. I am curious about the relationship between group polarization, political polarization, and collective action in contexts that have limited or no access to social media. The authors allude to this possibility when they discuss how Egypt cut off access to internet (see page 16), however, in this context, social media did still play an important role in propelling initial calls to collective action. Do similar processes/links between group polarization, political polarization and collective action exist in absence of SNS, or is it something unique about

social media that strengthens these connections? I am not necessarily recommending that the authors spend a lot of time on this in the paper, but I find it important to discuss. Relatedly, SNS do not exist in isolation.

To clarify, we now make the following three points on pages 13-14:

(1) Group polarization occurs through communication (Table 1; Table 2, Proposition 1). This communication is necessary for people to collectivize over social issues. This is necessary even for people who engage in collective action alone, but take “collective” action because it is on behalf of their group³⁴. For these actors, the communication may take place in the form of reading and internalising the arguments and opinions of likeminded others online. This is why some repressive regimes cut off access to SNSs to prevent “rebellion”; and some regimes implement offline curfews to prevent people meeting and communicating in person (Table 2, Proposition 5). Limited or no interaction can prevent social change³⁵.

(2) The affordances of SNSs amplify and facilitate the processes of communication that lead to polarization. Before the internet and SNSs, the process of communication was slower/perhaps smaller in scale, but substantively the same, and communication still facilitated collective action³⁵. For example, during the Protestant Reformation, Luther and his supporters leveraged the printing press to communicate their ideas. Therefore, communication offline versus via SNSs is similar, but SNSs include affordances that amplify the processes of offline communication, (sometimes) allowing for anonymity (see our response to Reviewer 1 #3), increasing reach, allowing many people to talk simultaneously, faster, and on a much larger scale; access to information is more instant, and recommendation algorithms make encountering likeminded others and material more likely³⁶.

(3) We agree that SNSs do not exist in isolation - people cross-post, using multiple networked applications, sites, and forums, as well as engaging in online communication whilst simultaneously communicating in multiple offline modes. This occurs during collective action³⁷. According to media multiplexity theory³⁸, the number of communication modes people use is positively related to the strength of the relationship between them and the people they are communicating with, suggesting that using SNSs to communicate alongside other modes of communication may strengthen an ingroup, better equipping them to act collectively. At the same time, offline events and communications shape and influence SNS interactions, and vice versa.

3. I appreciate how the authors list the different definitions of polarization and clarify that they refer to political polarization as “opinion-based” (see page 9). However, the other definitions of polarization are also particularly relevant to the arguments made. For instance, typically opinion-based/issue-based polarization can give way to affective polarization. Likewise, collective action can be fueled by co-radicalization, which is likely also fueled by group polarization. Focusing on one type of polarization has the potential to limit theoretical impact of the current manuscript. I recommend either providing stronger justification for why issue-based polarization is the specific polarization of focus or integrating the other types of polarization into the model.

Reviewer 3 notes that “...typically opinion-based/issue-based polarization can give way to affective polarization.” This is in line with an argument made by Bliuc et al. (2021), so we have now integrated this point in the revised text on pages 7-8 where we say:

“...as Jost and colleagues have explained¹, different types of polarization can affect each other. For instance, issue-based polarization is often a driver of affective polarization³⁹, as in cases where dissent over a newly emerged issue in society (e.g., the COVID-19 pandemic) polarizes the public into opposing groups, which then develop hostile attitudes

towards each other (affective polarization). In this sense, political polarization is best conceptualized as a process incorporating both issue and affective polarization, with issue-driven dissent creating conditions for issue-driven polarization, while affective polarization being its manifestation (or the attitude - behavioral component within this process⁴⁰).

In relation to the next point made by Reviewer 2 on collective action being "fueled by co-radicalization, which is likely also fueled by group polarization", we address it on pp. 7-8 where we say:

"Similarly, the inter-connections between different forms of polarization are clear in cases where collective action occurs in response to opposing groups becoming more extreme through exposure to each other's ideology and actions (co-radicalization; Table 1)."

To clarify, we don't focus on issue based polarization over affective polarization. Rather, we call for clearer theorizing on these interconnections, and focus on group polarization, which is the vehicle by which people consensualize over issues and associated emotions. Like EMSICA, we argue that they are integrated through discussion to form the "content" of new group identities (see our response to Reviewer 1, #4). We now clarify this on p. 15.

Reviewer #3:

Thanks for the opportunity to review this work. I really enjoyed reading this manuscript and found a lot to appreciate about it. The paper is on a very interesting and timely topic, and the integration of both the political polarization and group polarization literatures is a welcome addition as these are often situated in different areas when there are valuable synergies between these areas of study. I also found the paper very nicely written keeping the reader engaged throughout. I only have two comments about the paper in its current form that the authors should be able to address in a revision.

1. First, while I really like the idea to present the work as taking a balanced approach to examining the effects of polarization, there are aspects of the current framing that may be too simplistically valenced. For example, in the early introduction, the promotion of collective action is framed as if it is inherently good, while collective action can lead to positive outcomes or disastrous ones. Even movements that may have seemingly good ideals and goals can turn toxic and destructive through polarization. For example, despite promises of greater collective welfare and giving power to peasants, polarization emerging from social movements in China and Cambodia led to death of millions. While seemingly extreme, the point is that social change emerging from collective action is not inherently good or bad. Anti-vax protests during the pandemic may have perceived their actions as positive for standing up against what they deemed to an authoritarian state, but it may have simultaneously increased risk of disease among the wider public and been destructive to others. Therefore, it would be better to further nuance the writing to clarify that polarization has the potential to bring people together to engage in collective action, but it is agnostic on whether such collective action can end up in destruction or human progress. I would argue that while I see what the authors are getting at here about its potential benefits, there is a risk that polarization undermines social cohesion and cuts people off from each other making it difficult to perspective take and understand each other in a liberal democracy with a pluralistic populace. However, I understand what the authors are meaning to contribute to

the literature and suggest nuancing the writing without an insinuation of the valenced nature of social change emerging from collective action.

Please see our response to Reviewer 1, #8. We agree with Reviewer 3, and in the revised manuscript we have nuanced our stance on the valence of collective action, and on extremization (rather than extremism). Hopefully, taken together our clarifications avoid the inadvertent potential implication of our arguments that all collective action - of which violent extremism is one form - is inherently good.

2. On a second, more minor point, the framing of page 23 about RWP leaders attacking the consensus that criticizing minority members as racist is difficult to understand. What types of criticism of minority groups are racist? Any criticism? While there has been a concept creep (see Haslam et al., 2020; Haslam, 2016) over the meaning of racist, this seems like an odd choice of words. I write this review in the midst of a global conflict that has led to a spike in antisemitism worldwide, and some argue that almost any criticism of Israel as antisemitic as the sole Jewish state in the world, while others see it as entirely justified, so who gets to decide what constitutes free speech and what is hate speech? The larger point is that what is deemed as bigotry is in the eyes of the beholder.

We have now clarified the point we were making on p. 23 of the original manuscript (by referring to strategies used in right-wing discourse to justify and normalize "divisive and discriminatory" attitudes (so that they become mainstream rather than fringe in society). See p. 26 of revised manuscript, where we explain,

"First, RWP leaders reframed divisive and discriminatory rhetoric as free speech. That is, positions previously suppressed by laws and custom were shared and discussed, so that they became justified and normalized⁴¹, group polarization being therefore enabled (in line with Proposition 3)."

Besides these relatively fixable issues, I thought the paper read well and would make a useful contribution to the literature. I hope this work will encourage more empirical research on this useful topic of study.

References

- 1 Jost, J. T., Baldassarri, D. S. & Druckman, J. N. Cognitive–motivational mechanisms of political polarization in social-communicative contexts. *Nature Reviews Psychology* **1**, 560-576 (2022). <https://doi.org/10.1038/s44159-022-00093-5>
- 2 Center, P. R. Political Polarization in the American Public: How Increasing Ideological Uniformity and Partisan Antipathy Affect Politics, Compromise and Everyday Life. (2014).
- 3 Postmes, T., Haslam, S. A. & Swaab, R. I. Social influence in small groups: An interactive model of identity formation. *European Review of Social Psychology* **16**, 1-42 (2005). <https://doi.org/10.1080/10463280440000062>
- 4 Postmes, T., Spears, R., Lee, A. T. & Novak, R. J. Individuality and social influence in groups: Inductive and deductive routes to group identity. *Journal of Personality and Social Psychology* **89**, 747-763 (2005). <https://doi.org/10.1037/0022-3514.89.5.747>
- 5 Lüders, A., Carpentras, D. & Quayle, M. Attitude networks as intergroup realities: Using network-modelling to research attitude-identity relationships in polarized political contexts. *British Journal of Social Psychology* **n/a** (2023). <https://doi.org/https://doi.org/10.1111/bjso.12665>
- 6 Dinkelberg, A., O'Sullivan, D. J., Quayle, M. & Maccarron, P. Detecting opinion-based groups and polarization in survey-based attitude networks and estimating question relevance. *Advances in Complex Systems* **24**, 2150006 (2021). <https://doi.org/10.1142/S0219525921500065>
- 7 Turner, J. C., Oakes, P. J., Haslam, S. A. & McGarty, C. Self and Collective: Cognition and Social Context. *Personality and Social Psychology Bulletin* **20**, 454-463 (1994). <https://doi.org/10.1177/0146167294205002>
- 8 O'Reilly, C., Maher, P. J., Lüders, A. & Quayle, M. Sharing is caring: How sharing opinions online can connect people into groups and foster identification. *Acta Psychologica* **230**, 103751 (2022). <https://doi.org/https://doi.org/10.1016/j.actpsy.2022.103751>
- 9 Smith, L. G. E., Thomas, E. F. & McGarty, C. "We must be the change we want to see in the world": Integrating norms and identities through social interaction. *Political Psychology* **36**, 543-557 (2015). <https://doi.org/10.1111/pops.12180>
- 10 Bliuc, A. *et al.* Public division about climate change rooted in conflicting socio-political identities. *Nature Climate Change* **5**, 226 - 229 (2015).
- 11 McGarty, C., Bliuc, A. M., Thomas, E. F. & Bongiorno, R. Collective Action as the Material Expression of Opinion-Based Group Membership. *Journal of Social Issues* **65**, 839-857 (2009). <https://doi.org/10.1111/j.1540-4560.2009.01627.x>
- 12 Bliuc, A., McGarty, C., Reynolds, K. & Muntele, D. Opinion-based group membership as a predictor of commitment to political action. *European Journal of Social Psychology* **37**, 19-32 (2007). <https://doi.org/10.1002/ejsp.334>
- 13 Bliuc, A.-M., Betts, J., Vergani, M., Iqbal, M. & Dunn, K. Collective identity changes in far-right online communities: The role of offline intergroup conflict. *New Media & Society* **21**, 1770-1786 (2019).
- 14 Williams, H. T. P., McMurray, J. R., Kurz, T. & Lambert, F. H. Network analysis reveals open forums and echo chambers in social media discussions of climate change. *Global Environmental Change-Human and Policy Dimensions* **32**, 126-138 (2015). <https://doi.org/10.1016/j.gloenvcha.2015.03.006>
- 15 Moscovici, S. & Zavalloni, M. The group as a polarizer of attitudes. *Journal of Personality and Social Psychology* **12**, 125-135 (1969).
- 16 O'Reilly, C., Maher, P. J., Smith, E. M., MacCarron, P. & Quayle, M. Social identity emergence in attitude interactions and the identity strengthening effects of cumulative attitude agreement. *European Journal of Social Psychology* **n/a** (2023). <https://doi.org/https://doi.org/10.1002/ejsp.3000>

- 17 Turner, J. C., Wetherell, M. S. & Hogg, M. A. Referent informational influence and group polarization. *British Journal of Social Psychology* **28**, 135-147 (1989).
- 18 McGarty, C., Turner, J. C., Hogg, M. A., David, B. & Wetherell, M. S. Group polarization as conformity to the prototypical group member. *British Journal of Social Psychology* **31**, 1-20 (1992).
- 19 Oakes, P. J., Haslam, S. A., Morrison, B. & Grace, D. Becoming an ingroup: Reexamining the impact of familiarity on perceptions of group homogeneity. *Social Psychology Quarterly* **58**, 52-61 (1995).
- 20 Myers, D. G. & Bishop, G. D. Discussion effects on racial attitudes. *Science* **169**, 778-779 (1970).
- 21 Koudenburg, N., Kutlaca, M. & Kuppens, T. The experience and emergence of attitudinal consensus in conversations. *European journal of social psychology* (2023). <https://doi.org/10.1002/ejsp.2992>
- 22 Turner, J. C., Hogg, M. A., Oakes, P. J., Reicher, S. D. & Wetherell, M. S. *Rediscovering the social group: A self-categorization theory.*, (Blackwell, 1987).
- 23 McGarty, C. & Penny, R. E. C. Categorization, accentuation and social judgement. *British journal of social psychology* **27**, 147-157 (1988). <https://doi.org/10.1111/j.2044-8309.1988.tb00813.x>
- 24 Reicher, S., Spears, R. & Postmes, T. A social identity model of deindividuation phenomena. *European Review of Social Psychology* **6**, 161-198 (1995).
- 25 Spears, R., Lea, M., Corneliussen, R. A., Postmes, T. & Ter Haar, W. Computer-mediated communication as a channel for social resistance: The strategic side of SIDE. *Small Group Research* **33**, 555-574 (2002).
- 26 Thomas, E. F., Mavor, K. I. & McGarty, C. Social identities facilitate and encapsulate action-relevant constructs: A test of the social identity model of collective action. *Group Processes & Intergroup Relations* **15**, 75-88 (2012). <https://doi.org/10.1177/1368430211413619>
- 27 Thomas, E. F., McGarty, C. & Mavor, K. I. Aligning identities, emotions, and beliefs to create commitment to sustainable social and political action. *Pers. Soc. Psychol. Rev.* **13**, 194-218 (2009). <https://doi.org/10.1177/1088868309341563>
- 28 Tajfel, H. & Turner, J. C. in *The social psychology of intergroup relations* (eds S. Worchel & W. G. Austin) (Brooks/Cole, 1979).
- 29 Wetherell, M. S. in *Rediscovering the social group: A self-categorization theory.* (eds J.C. Turner *et al.*) 142-170 (Blackwell, 1987).
- 30 Hopkins, N. & Kahani-Hopkins, V. Reconceptualizing extremism and moderation: From categories of analysis to categories of practice in the construction of collective identity. *British journal of social psychology* **48**, 99-113 (2009). <https://doi.org/10.1348/014466608X284425>
- 31 Iyengar, S., Lelkes, Y., Levendusky, M., Malhotra, N. & Westwood, S. J. The Origins and Consequences of Affective Polarization in the United States. *Annual Review of Political Science* **22**, 129-146 (2019). <https://doi.org/10.1146/annurev-polisci-051117-073034>
- 32 Lamm, H., Trommsdorff, G. & Rost-Schaude, E. Group-Induced Extremization: Review of Evidence and a Minority-Change Explanation. *Psychol. Rep.* **33**, 471-484 (1973). <https://doi.org/10.2466/pr0.1973.33.2.471>
- 33 Smith, L. G. E., Blackwood, L. & Thomas, E. F. The Need to Refocus on the Group as the Site of Radicalization. *Perspectives on Psychological Science* **15**, 327-352 (2020). <https://doi.org/10.1177/1745691619885870>
- 34 Wright, S. C., Taylor, D. M. & Moghaddam, F. M. Responding to membership in a disadvantaged group: From acceptance to collective protest. *Journal of Personality and Social Psychology* **58**, 994-1003 (1990). <https://doi.org/10.1037/0022-3514.58.6.994>
- 35 McGarty, C., Thomas, E. F., Lala, G., Smith, L. G. E. & Bliuc, A. New technologies, new identities, and the growth of mass opposition in the Arab Spring. *Political Psychology* **35**,

- 725-740 (2014). <https://doi.org:10.1111/pops.12060>
- 36 Brown, O., Smith, L. G. E., Davidson, B. I. & Ellis, D. A. The problem with the internet: An affordance-based approach for psychological research on networked technologies. *Acta Psychologica* **228**, 103650 (2022).
<https://doi.org:https://doi.org/10.1016/j.actpsy.2022.103650>
- 37 Smith, L. G. E., Piwek, L., Hinds, J., Brown, O. & Joinson, A. Digital traces of offline mobilization. *Journal of Personality and Social Psychology*, No Pagination Specified-No Pagination Specified (2023). <https://doi.org:10.1037/pspa0000338>
- 38 Haythornthwaite, C. Social networks and Internet connectivity effects. *Information, Communication & Society* **8**, 125-147 (2005). <https://doi.org:10.1080/13691180500146185>
- 39 Betts, J. M. & Bliuc, A. M. in *Proceedings of the 2022 Winter Simulation Conference, WSC 2022*. (eds B. Feng *et al.*) 370-381 (IEEE, Institute of Electrical and Electronics Engineers).
- 40 Bliuc, A.-M., Bouguettaya, A. & Felise, K. D. Online Intergroup Polarization Across Political Fault Lines: An Integrative Review. *Frontiers in psychology* **12**, 641215-641215 (2021).
<https://doi.org:10.3389/fpsyg.2021.641215>
- 41 Faulkner, N. & Bliuc, A.-M. 'It's okay to be racist': moral disengagement in online discussions of racist incidents in Australia. *Ethnic and racial studies* **39**, 2545-2563 (2016).
<https://doi.org:10.1080/01419870.2016.1171370>

27th Feb 24

Dear Professor Smith,

Your Perspective titled "Polarization is the Psychological Foundation of Collective Engagement" has now been seen by one reviewer (Reviewer #3), whose comments appear below. Based on our evaluation and in light of their advice I am delighted to say that we are happy, in principle, to publish it in Communications Psychology under a Creative Commons 'CC BY' open access license.

If you revise the paper to meet the editorial requests detailed below and on the supporting documents, and if the revised paper is in Communications Psychology format, in an accessible style, and of appropriate length, we shall accept it for publication immediately.

EDITORIAL REQUESTS:

Reviewer #2 raised an important point in their previous report (point #2) to which you replied at length in your rebuttal. However, we ask that you also incorporate a briefer version of the explanation in the manuscript, for the benefit of future readers who may have similar thoughts as the referee but won't know unprompted about the detailed explanation in the transparent peer review file.

Moreover, as you implement the final revisions we ask you to pay particular attention to two requests. First, we ask that you review how references are used within sentences. It is currently often unclear which part of a complex argument, or a list of statements, is supported by a given reference. You will find some pointers regarding the issue on the attached manuscript. Second, although the use of real-world examples is a strength of your work, as currently presented, it also sets the text at risk of becoming less rather than more comprehensible to future readers (for example, a Ph.D. student in 2034, born in 2012, might struggle to recognize the many events from the 2010s that are referred to offhand). Please provide sufficient explanation and references to all real-world examples to ensure the timeliness of the piece does not work against its readability in future.

* Please review the changes in the attached copy of your manuscript, which has been edited for style, and address the comments and queries I have added.

*Please review our specific editorial comments and requests regarding your manuscript in the attached "Editorial Requests Table". Please outline your response to each request in the right-hand column. Please upload the completed table with your manuscript files as a Related Manuscript file.

[Link redacted]

We hope to hear from you within two weeks; please let us know if the process may take longer. As Marike Schiffer will handle your work from this point forward, please direct all inquiries to her.

Best regards,

Antonia Eisenkoeck & Marike

Antonia Eisenkoeck
Senior Editor
Communications Psychology

Marika Schiffer, PhD
Chief Editor
Communications Psychology

REVIEWERS' COMMENTS:

Reviewer #3 (Remarks to the Author):

I was one of the original reviewers of the paper. The authors have done a great job addressing my concerns and comments. Congrats on an interesting paper!